# Compositionally-distinct ultra-low velocity zones on Earth's core-mantle boundary

Mingming Li [1], Allen K. McNamara[2], Edward J. Garnero[1] & Shule Yu[1]

The Earth's lowermost mantle large low velocity provinces are accompanied by small-scale ultralow velocity zones in localized regions on the core-mantle boundary. Large low velocity provinces are hypothesized to be caused by large-scale compositional heterogeneity (i.e., thermochemical piles). The origin of ultralow velocity zones, however, remains elusive. Here we perform three-dimensional geodynamical calculations to show that the current locations and shapes of ultralow velocity zones are related to their cause. We find that the hottest lowermost mantle regions are commonly located well within the interiors of thermochemical piles. In contrast, accumulations of ultradense compositionally distinct material occur as discontinuous patches along the margins of thermochemical piles and have asymmetrical cross-sectional shape. Furthermore, the lateral morphology of these patches provides insight into mantle flow directions and long-term stability. The global distribution and large variations of morphology of ultralow velocity zones validate a compositionally distinct origin for most ultralow velocity zones.

[1] Arizona State University, School of Earth and Space Exploration, PO Box 871404, Tempe, AZ 85287-1404, USA. [2] Michigan State University, Department of Earth and Environmental Sciences, Natural Science Building, East Lansing, MI 48824, USA. Correspondence and requests for materials should be addressed to M.L. (email: Mingming.Li@asu.edu)

Ultralow velocity zones (ULVZs) are mapped as geographically isolated zones of seismic anomalies detected on the core-mantle boundary (CMB)[1, 2] with a significant reduction of seismic velocity (up to 10% for P-wave and 30% for S-wave velocities) and sometimes increased density[3, 4]. Mapped as thin (5–40 km) and relatively small (e.g., hundreds of kilometer laterally, or less[2, 4–6], but sometimes up to 1000 km long[7, 8]), ULVZs are more commonly found within or near the large low velocity provinces (LLVPs)[9]. For this reason, along with the predominance of the S-velocity (Vs) drops being up to three times that of P-velocity (Vp) reductions, the ultralow wave speeds have been attributed to partial melt due to being in the hottest lowermost mantle regions[2, 4].

At odds with the solely partial melt hypothesis is that some seismic studies identify ULVZs well outside of the seismically observed LLVPs[10–13] including beneath subduction regions[9], where temperatures are assumed to be far lower than in the presumed upwelling regions of LLVPs. Furthermore, some ULVZs do not have Vs reduction substantially greater than their Vp reduction[14]. Hypotheses other than solely partial melt may thus be necessary. A number of hypotheses have been proposed, including iron-enriched $(Mg,Fe)O$[15, 16], iron-enriched post-perovskite[17] (recent geodynamic modeling results show that patches of post-perovskite can be temporarily stable within LLVPs[18]), subducted banded iron formations[19], subducted oceanic crust[20] or other slab-derived materials[21, 22], and products of chemical reactions between the silicate mantle and Fe-rich core[23, 24]. While these possibilities (as well as partial melt) may all

be viable, their relationship to deep mantle flow, especially in regards to being swept towards upwelling regions and their geometrical relationship to LLVPs remains unknown. Of particular interest is the thermochemical pile hypothesis to explain LLVPs, whereby dense basal material is swept into piles to explain the seismically observed LLVPs[25–27]. Two fundamental questions are: where are the highest temperatures inside LLVPs, and are they different from accumulation locations of any additional, ultradense material that may reside at the base of the mantle? Understanding the dynamics, destinations, and morphologies of ULVZs caused by a compositionally distinct vs. partial melt origin is necessary to provide a meaningful framework for the distribution of seismic observations. We thus carried out very high resolution, three-dimensional thermochemical numerical convection calculations to study the distribution and morphology of ULVZs.

Here we explore ULVZs attributed to ultradense, compositionally distinct material, as well as ULVZs attributed to melting in the hottest deep mantle. We find that the hottest lowermost mantle regions, where partial melting could occur to explain ULVZs, are located well within the interiors of thermochemical piles. In contrast, accumulation of ultradense compositionally distinct material occurs as discontinuous patches along the margins of thermochemical piles and have an asymmetrical cross-sectional shape. The origin of ULVZs, therefore, can be constrained from their locations and shapes. The global distribution and large variations of morphology of the seismologically observed ULVZs indicate a compositionally distinct

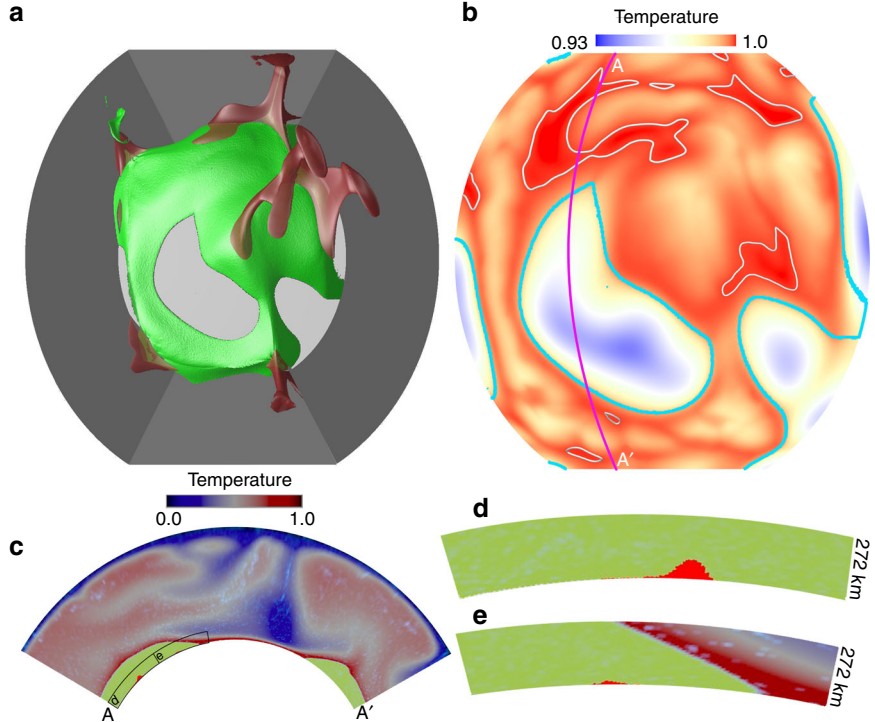

**Fig. 1** Morphology and distribution of ULVZs caused by partial melting. **a** *Top view* of the model with mantle plumes (*red isosurface* of temperature of 0.683) forming on tops of thermochemical piles (*green isosurface*). The *gray side boundaries* define the domain of the 3D partial spherical geometry of the model, which spans 120° in both longitudinal and colatitudinal directions. The downwellings (not shown) sweep pile material away, exposing the core (*light gray*) in these regions. **b** Temperature field in map-view at 5 km above the CMB. *Light gray contours* at $T = 0.999$ show the hottest 10% regions of the piles by area. *Thick cyan lines* show the edges of piles. **c** Cross-section at locations marked by magenta line in Fig. 1b. The temperature field is shown by *dark blue* to *red color*, the thermochemical piles are shown in *green color* in the lowermost mantle, and the *small red patches* within the piles at the *bottom* of the mantle show the hottest regions. **d**, **e** Zoomed-in at the regions outlined by *black boxes* in Fig. 1c. The hottest regions (*red patches* inside the pile) are candidate regions for melting, and we identify the hottest regions where temperature is higher than mantle solidus. At 5 km above CMB, the hottest 10% pile regions by area are identified as candidate regions for melting by assuming a mantle solidus of $T = 0.999$ (non-dimensional) at this depth. For other depths, the mantle solidus decreases with decreasing depth with a rate of ~0.8 K km$^{-1}$[38]. All panels **a**–**e** are shown at 218 Myr

origin for most ULVZs, and that Earth's lowermost mantle contains small-scale compositional heterogeneities with elevated intrinsic density. ULVZs within LLVPs, however, might be explained by partial melting alone.

## Results

**Description of mantle convection models**. Our reference model includes thermochemical piles, motivated by the multiple lines of evidence arguing a chemically distinct origin of LLLVPs[28–35]. The conservation equations of mass, momentum and energy are solved using our modified version of the code CitcomCU[36] in the Boussinesq approximation (Methods). We employ a Rayleigh number $Ra = 9.8 \times 10^6$ for most cases (Supplementary Table 1, using mantle thickness as the length-scale for non-dimensionalization). A 50× viscosity increase is employed from the upper mantle to the lower mantle (Supplementary Fig. 1). The temperature-dependent part of the viscosity is expressed as $\eta_T = \exp[A(0.6 - T)]$, where $T$ is non-dimensional temperature, and we use a non-dimensional activation coefficient of $A = 9.21$ for most cases (Supplementary Table 1), leading to a 10000× viscosity range across the mantle due to changes in temperature. We employ a three-dimensional, partial-sphere geometry (Fig. 1a) in which the longitude and colatitude span 120°, and the dimensionless radius ranges from 0.55 to 1.0 (thus, from the CMB to the surface). We utilize 512, 512 and 128 elements in longitudinal, colatitudinal, and radial directions, respectively. The mesh is refined with depth resulting in a resolution of 5 km radially and ~14.5 km laterally near the CMB. All boundaries are free-slip, isothermal at top and bottom, and insulating along the sides. The models are heated both from below and internally with a non-dimensional heat production rate of $H = 60$ (using Earth's radius as the length-scale for non-dimensionalization).

We developed a hybrid tracer scheme to track composition (Methods), that simultaneously employs both ratio and absolute tracing methods[37]. The background mantle and the thermochemical piles are modeled with ~710 million ratio tracers and the ultradense ULVZ material is modeled with ~ 50–110 million absolute tracers, depending on the volume of ULVZ material (Supplementary Table 1). The hybrid tracer method more efficiently computes the advection of multi-scale composition, including both large-scale thermochemical piles and much smaller-scale accumulations of ultradense materials. The intrinsic density anomaly ($\Delta\rho$) of each compositional component is non-dimensionalized as compositional buoyancy number $B$. The effective intrinsic density of each element is calculated by averaging the densities of each component, leading to an "effective buoyancy ratio", $B^{eff}$. To construct an initial condition, we carry out a calculation with two compositional components (background mantle and thermochemical pile material) and we use the quasi-steady state temperature and composition field as initial condition for models in this study.

We perform 2 types of experiments, both of which include thermochemical piles to represent LLVPs. In the first set of experiments, we explore the positions and shapes of ULVZs caused by partial melting in the hottest mantle regions. In other words, we examine the morphology of the hottest lowermost mantle regions. In the second set of experiments, we explore the positions and shapes of ULVZs caused by the accumulation of the ultradense compositional component. We then examine the morphology of these accumulations.

**ULVZs caused purely by partial melting**. We first examine the locations of ULVZs due to partial melting alone (Case 1). The amount of partial melting above the CMB is controlled by the solidus temperature, liquidus temperature, and the mantle temperature above the CMB. The solidus temperature and liquidus temperature of a synthetic sample with chondritic-type composition at CMB pressure were measured by previous mineral physics experiments to be ~4150 and ~4725 K[38], respectively, and the solidus and liquidus temperatures for a natural fertile peridotite at CMB pressure were measured to be ~4180 and 5375 K, respectively[39]. However, the solidus temperature for a pyrolitic composition with ~400 p.p.m. $H_2O$ has been reported to be as low as ~3570 K[40]. Largely due to our limited knowledge about the lowermost mantle composition such as the amount of $H_2O$, the solidus temperature and liquidus temperature near the CMB pressure are not well constrained. In addition, there is large uncertainty of the CMB temperature, which has been suggested to be in the range of from ~2500–2800 K to ~3300–4300 K[41], and the temperature of the thermal boundary layer above the CMB is poorly constrained. Our geodynamic models are non-dimensionalized and therefore do not independently constrain the absolute value of dimensional temperature. To convert non-dimensional to dimension temperature requires a choice for CMB temperature, which is not well constrained by observations. Because of these uncertainties, it becomes impractical to determine the amount of partial melting above the CMB in our models by comparing the dimensional lowermost mantle temperature in our models with the solidus and liquidus temperature at the CMB pressure measured in previous mineral physics experiments. Nonetheless, if there are ULVZs above the CMB caused by partial melting alone, they most likely exist in the hottest regions in the lowermost mantle. We thus focus on examining the location of hottest regions in the lowermost mantle in Case 1.

Case 1 includes 2 compositions: background mantle and piles with a buoyancy number $B_p = 0.8$ (or 3.6% denser than the background mantle if scaled using reference temperature and thermal expansivity as given in Supplementary Table 2). Figure 1 shows a snapshot at 218 Myr. In this study, the geological time is scaled by the transit time and we assume that one transit time (the time it takes for a slab to descend from surface to the CMB) equals to 60 Myrs[42, 43]. Figure 1a illustrates thermochemical piles (*green*) with mantle plumes (*red*) rising from cusps along their tops. Figure 1b shows the temperature field at 5 km height above the CMB, in which it is observed that the hottest 10% regions of the piles by area (marked by light gray contours of $T = 0.999$) occur within pile interiors, well inward from their edges (pile edges are outlined by *cyan lines*). The dimensional temperature for the hottest 10% regions at this depth is in the range of ~3600–4000 K, if dimensionalized with a reference potential temperature of $\Delta T = 2500$ K (Supplementary Table 2), after adding an adiabatic temperature increase from the surface (with a temperature of 273 K) down to 5 km above the CMB with an adiabatic thermal gradient of 0.3–0.4 K km$^{-1}$. Interestingly, these hottest regions have a temperature comparable to the solidus temperature of a pyrolitic or chondritic composition near the CMB pressure, depending on the $H_2O$ content in the lowermost mantle[38–40]. However, it needs to be emphasized that the dimensional temperature in our models depends on the choice of reference temperature for scaling. We thus focus on the location of hottest regions that are the best candidate locations for partial melting.

We plot the hottest regions in the cross-section shown in Fig. 1c, with zoom-ins shown in Fig. 1d, e. The hottest regions reside within and with some distance from the pile edges because of cooling of thermochemical pile margins by the cooler, surrounding non-pile mantle (Supplementary Note 1; Supplementary Fig. 2). We find the hottest regions occur well within the interior of thermochemical piles throughout the model run

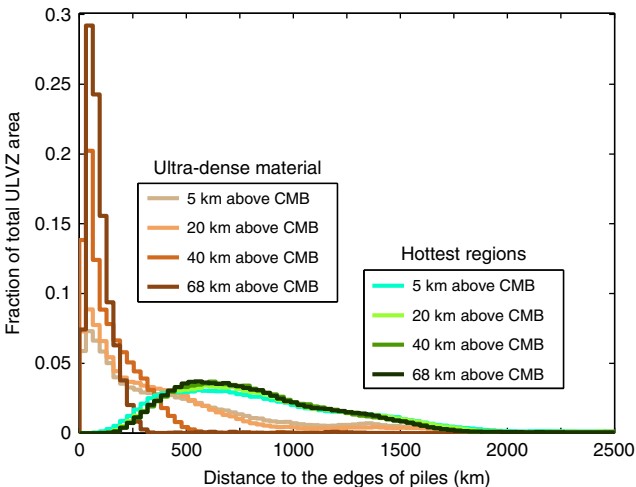

**Fig. 2** Distance of ULVZs from thermochemical pile edges. For each distance in the horizontal axis, the corresponding vertical axis shows the areal fraction of hottest regions or accumulations of ultradense material at this distance. Here, the hottest regions refer to candidate regions for melting (e.g., Fig. 1b–e), the regions with accumulations of ultradense material refer to places where the effective buoyancy ratio ($B^{eff}$) is larger than 1.5, and the edges of piles are identified at locations with $B^{eff} = 0.1$. The distances are calculated at 4 depths of 5, 20, 40km and 68 km above the CMB

(Supplementary Movie 1). We compute the lateral distance of hottest regions from the closest edges of thermochemical piles throughout the model run. We exclude the hottest regions that are within 500 km from side boundaries of the model domain. We find that the distances between the hottest regions and the edges of thermochemical piles range from a minimum of 100's km to over 1500 km, with a peak at around 500–1000 km (Fig. 2).

**ULVZs caused by ultradense material**. A second set of experiments considers the dynamics and evolution of ultradense material as a cause of ULVZs (Case 2). These experiments have three compositions: background mantle, thermochemical piles, and a small volume of ultradense material with a buoyancy number of $B_u = 2.0$ (or 9% denser than background mantle if scaled using reference parameters given in Supplementary Table 2). The ultradense material is initially introduced as a ubiquitous uniform layer in the lowermost 5 km of the mantle, and it quickly advects toward the pile edges, accumulating into discontinuous patches of varying size and shape (Supplementary Movie 2). Figure 3 shows a snapshot of this case at 227 Myr. Figure 3a displays the distribution of ultradense ULVZ material (*red isosurfaces*) underneath the thermochemical piles (partially transparent *green isosurfaces*). The accumulations vary in size from ~100 to ~1000 km across and ~ 5–100 km thick and have either rounded or linear map-view morphologies (discussed later). An interesting point to note is that the accumulations form into discontinuous patches, as opposed to ubiquitous, continuous ribbons along pile edges implied from 2D studies[9]. Figure 3b is a zoom-in of Fig. 3a that displays the effective buoyancy ratio in regions with ultradense material 5 km above the CMB, illustrating the heterogeneity of density within the accumulations, caused by stirring with the surrounding mantle. Figure 3c demonstrates that accumulations of ultradense material are typically quite thin, except for small regions within particularly large accumulations, where local heights may reach up to 100 km above the CMB. Figure 3d–f illustrates the variability in cross-sectional shape of the accumulations.

The lateral width of the accumulations greatly varies from place to place, and the cross-sectional shape of the accumulations is asymmetrical, thicker on the side in contact with the background mantle. This asymmetrical shape is due to differential viscous coupling, as noted in a previous 2D study[9].

Figure 4 shows the compositional field at 5 km above the CMB for a time sequence of snapshots for Case 2, illustrating the time-dependence of the distribution of ultradense material. At 121 Myr (Fig. 4a), two large patches of ultradense material (labeled U1 and U2) are located at the edge of the pile. At 160 Myr (Fig. 4b), U2 has been advected into a linear shape, whereas U1 has maintained its rounded shape. At 227 Myr (Fig. 4c), U2 has split into three parts: a remnant of U2 (still labeled as U2) migrated toward U1, another formed into a smaller accumulation with relatively rounder shape (labeled U3), and another had been entrained into the pile, up along its side, and back down again (U4). In Fig. 4c, the ultradense material in U2 and U4 has experienced higher degree of stirring with pile material, leading to a lower effective buoyancy ratio (i.e., effective intrinsic density) in these patches than U1 and U3. In general, we observe that regions of long-term, stable, horizontally convergent mantle flow produces longer-lived, rounded accumulations of ultradense material. In contrast, linear accumulations are the result of ultradense material on the move, toward a location of more-stable convergent flow. Thus, ULVZ shape can change over time scales as short as tens of Myr.

Similar to Case 1, we compute the lateral distance of regions with accumulations of ultradense material from the closest edges of thermochemical piles for Case 2 (Fig. 2). We compute the distances throughout the model run but we exclude the first 50 Myr for Case 2 when the initial global layer of ultradense material is advecting to the edges of piles. We also exclude the regions with accumulations of ultradense material that are within 500 km from side boundaries of the model domain. In contrast to the wide range of distances between hottest regions and pile edges, the compositionally distinct ultradense material generally accumulates along the edges of thermochemical piles. At depths of 40 and 68 km above the CMB, the majority of ultradense materials occurs within ~300 km from the pile edges (Fig. 2). At depths of 5 and 20 km above the CMB, the patches of ultradense material become much larger than at shallower depths (Fig. 3d), which leads to a significant amount of ultradense materials occurring between ~ 300–800 km from the pile edges. However, even at depths of 5 and 20 km above the CMB, the largest fraction of ultradense materials still occurs within ~300 km from pile edges.

**Results of other geodynamic models**. We also explored different combinations of parameters, including intrinsic density, volume and an intrinsic compositional viscosity decrease of the ultradense material, temperature-dependence of viscosity, Rayleigh number, and intrinsic density of thermochemical piles. Varying these parameters leads only to second-order differences from Case 1 or Case 2, but the fundamental conclusions about the distribution and accumulation of ultradense material, and the locations of hottest regions remain unchanged. We find that increasing the intrinsic density (Case 3) or initial volume (Case 4) of ultradense material acts to increase the size of the accumulations of ultradense material. Reducing the intrinsic density of ultradense material (Case 5) leads to more stirring between the ultradense material and pile material than Case 2. Using a larger activation coefficient ($A = 11.51$) for the temperature-dependent viscosity in Case 6 results in a slightly increase of the size of accumulations ultradense material. Similar to previous geodynamic modeling results[44, 45], we find that the morphology of

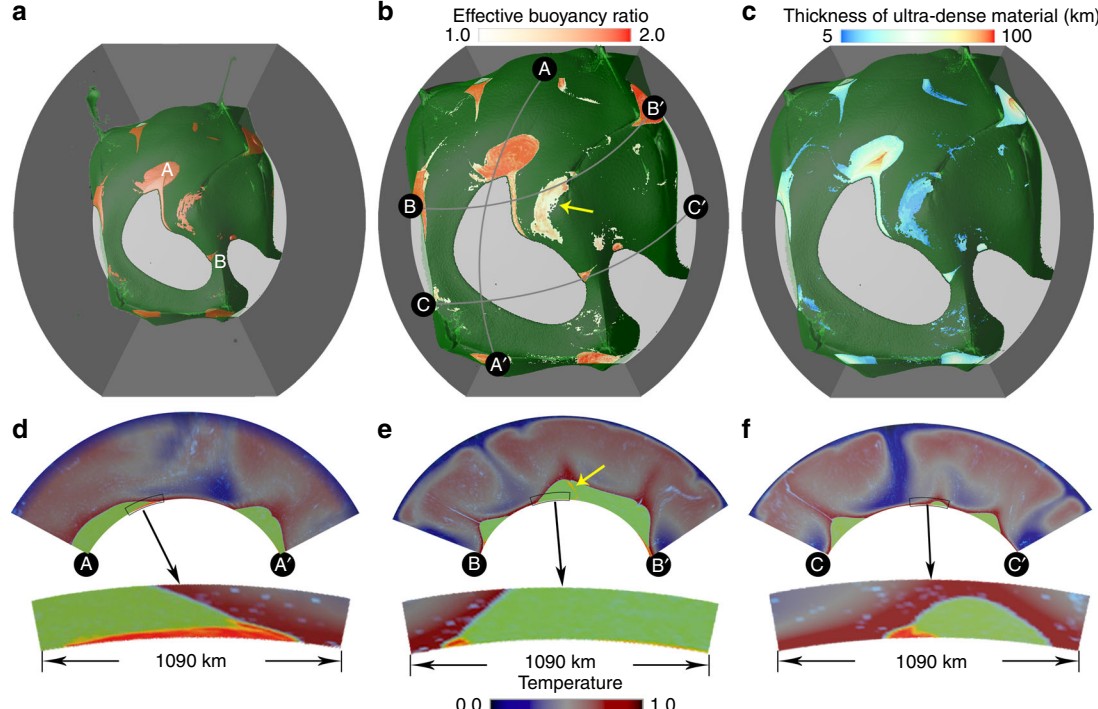

**Fig. 3** Morphology and distribution ULVZs caused by compositionally distinct ultradense material. **a** Distribution of ultradense material (*red isosurface*) beneath large-scale thermochemical piles (partially transparent *green isosurface*). The lateral length-scale of accumulations of ultradense material at locations A and B is ~1000 and ~100 km, respectively. The exposed Earth's core in downwelling regions is shown by *light gray color*. **b** A zoom-in of panel a along with the locations of 3 cross-sections. The effective buoyancy ratio field at 5 km above CMB show the locations of ultradense material (*white to red color*). The *yellow arrow* points to the region with relatively low-effective buoyancy ratio (*whitish color*) caused by stirring of ultradense material with pile material. **c** Same as in **b** but with the thickness of accumulations of ultradense material shown. **d**–**f** Cross-section at locations marked by gray lines in **b**. The temperature field is shown by *dark blue to red color*, the thermochemical piles are shown in *green color* in the lowermost mantle and the ultradense material is shown by *orange-red color* at the base of piles. The *yellow arrow* in **e** points to ultradense material that has been viscously entrained up to the top of the pile along the pile edges, and later sank to the bottom of the pile. Beneath each panel is a zoomed-in region outlined by the black box at the edges of piles. All panels **a**–**f** are shown at 227 Myr

thermochemical piles is affected by using a different Rayleigh number, temperature-dependence of viscosity, and intrinsic density of thermochemical piles (Cases 7–11). However, for all cases discussed above, the distribution of ultradense material is similar to that in Case 2, with the majority of ultradense materials forming into discontinuous patches at the edges of thermochemical piles (Supplementary Figs. 3–8), and the hottest lowermost mantle regions are generally located well within the interiors of thermochemical piles (Supplementary Figs. 9–10). A more detailed discussion about the modeling results for Cases 3–11 is provided in the Supplementary Note 2.

One caveat is that we do not include viscous dissipation in our models, so we do not have shear heating in the piles, which we consider negligible given their low viscosities. However, it is not inconceivable that certain combinations of material properties could lead to viscous heating, and therefore possible partial melting in other parts of the pile as well, such as near the edges where flow is changing direction.

**Comparison with seismic observations of ULVZs.** We show in Fig. 5a the seismic shear-wave tomography model S40RTS near the CMB[46], with the edges of LLVPs marked by orange contours. We plot observations of ULVZs together with the edges of LLVPs in Fig. 5b. Here, we only select studies of ULVZs using core reflected waves (ScS, PcP, ScP), in which the locations of ULVZs have minimum uncertainties (in comparison to the core waves or long path diffracted waves). The lateral size of the ULVZs is computed based on 1/4 wavelength Fresnel zones of the CMB

reflection location for the waves used in each study. A list of references for these ULVZ observations is provided in the Supplementary Table 3 and Supplementary References.

As shown in Fig. 5b and also summarized in the Fig. 1 of ref. [9], the ULVZs exhibit a variety of shapes and sizes, similar to the accumulations of ultradense material as labeled U1-4 in Fig. 4. For example, a larger-than-average, rounded ULVZ is observed near the north edges of the Pacific LLVP (Fig. 5b) and beneath Hawaii[7], not unlike the large rounded U1 (Fig. 4); a linear shape ULVZ detected in the SW Pacific[8] may be similar to the long linear U2 in Fig. 4b, c; and the ULVZs with small lateral-scale detected in many regions are analogous to the small U3. Our results show variable degrees of stirring of ultradense material with pile material (U2 and U4 compared to U1 and U3 in Fig. 4c), which may be analogous to the variable density increases observed in ULVZs[3, 4]. Our geodynamic modeling results also suggest that the accumulations of ultradense material are not ubiquitous along pile edges but form into discontinuous patches with variable morphology, demonstrating that not all LLVP margins are expected to contain ULVZs. This is supported by a recent detection of intermittent and unevenly distribution ULVZs at the northeastern margin of the Pacific LLVP[47].

Similar to that shown in Fig. 2, we compute the closest lateral distances of observed ULVZs (as shown in Fig. 5b) to the edges of LLVPs along the CMB (Methods). Figure 5c shows that 55.5% of the computed ULVZ area occurs outside of the LLVPs (denoted with negative distance) and 44.5% ULVZ area occurs within the LLVPs (denoted with positive distance). We find that most

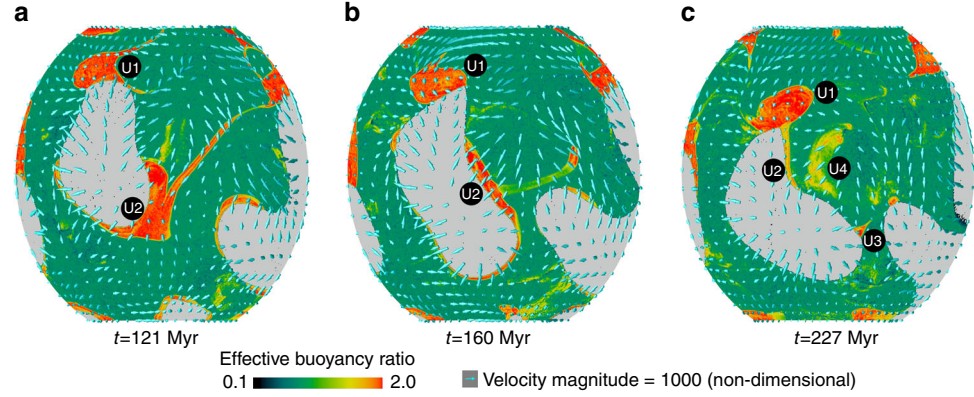

**Fig. 4** Time evolution of the location and morphology of ultradense material patches. **a–c** Composition field (represented by effective buoyancy ratio) at 5 km above the CMB showing locations of ultradense material (*reddish*) and pile material (*greenish*). The *yellowish color* shows a mixing of ultradense material with pile material. The *cyan arrows* show mantle flow velocity. U1, U2, U3, and U4 are markers that track the patches of ultradense material, as explained in the text. *Light gray* color represents Earth's core

ULVZs are in close proximity of LLVP margins (Fig. 5c). Outside of LLVPs, almost all the ULVZ area occurs within 800 km from the edges of LLVPs. Inside of LLVPs, we find that the ULVZ area decreases linearly with the increase of distance to LLVP edges, and there is no ULVZ area occurring more than 1200 km from the LLVP edges.

We calculated the distance of ULVZs to the edges of LLVPs for other five tomography models (Supplementary Fig. 11). The amount of ULVZ area outside and inside of LLVPs differs somewhat between models, since the locations of the LLVP edges slightly differ among models. However, the general conclusion holds that ULVZs are both outside and inside the LLVPs, with most ULVZ area occurring within ~800 km from LLVP edges of the tomographic models tested. The proximity of the observed ULVZs near LLVP edges is similar to the ultradense materials in our geodynamic models occurring near the edges of thermochemical piles (Fig. 2), suggesting a compositionally distinct component to ULVZs.

## Discussion

The seismically derived ULVZs studied here have variable lateral dimensions, morphologies, and locations, consistent with a compositionally distinct origin of ULVZs. For the Earth, the crystallization of basal magma oceanic may initially produce a thin layer of ultradense material on the CMB[29, 48]. The ultradense materials may be produced by the interaction between the core and the mantle and they may be produced at any location where the core and mantle interact[23, 24]. In addition, the subduction of slabs may bring some intrinsically dense materials to the lowermost mantle outside of the LLVPs[19–22]. Our experiments are geared toward understanding thermochemical convection at equilibrium conditions; however, our model setup also allows us to explore (in a limited manner) how ultradense material gets swept from the surrounding mantle to the edges of thermochemical piles. Because our initial condition consists of a thin, uniform ultradense layer ubiquitous along the CMB, the early times of the calculation exhibit the sweeping of this material toward the piles (Supplementary Movie 2; Supplementary Fig. 12). It demonstrates that any high-density compositional heterogeneity outside of piles is being advected toward the global upwelling regions (where the piles exist). Therefore, if ULVZs are caused by ultradense subduction remnants[19–22] or core-mantle boundary reaction products[23, 24], we expect to observe them outside of piles as they are being advected toward them. Note that

if compositional ULVZs have a lower solidus than background mantle, they may also include partial melt.

Though our results suggest that ULVZs located outside or at the edges of LLVPs are compositionally distinct from their surroundings, a small number of ULVZs located well within LLVPs[9] (Fig. 5b) may be caused solely by partial melting. Interestingly, partial melting within the LLVPs would likely alter composition[49], perhaps producing a source of intrinsically dense heterogeneity with lower melting temperature[49–51] that would continually advect toward LLVP edges.

## Methods

**Numerical modeling.** We perform high-resolution three-dimensional calculations to investigate the morphology, distribution and dynamics of ULVZs by solving the following non-dimensional equations for conservation of mass, momentum, and energy under the Boussinesq approximation:

$$\nabla \cdot \mathbf{u} = 0 \tag{1}$$

$$-\nabla P + \nabla \cdot (\eta \dot{\epsilon}) = \xi Ra \left( T - B^{\mathrm{eff}} \right) \mathbf{r} \tag{2}$$

$$\frac{\partial T}{\partial t} + (\mathbf{u} \cdot \nabla) T = \nabla^2 T + H \tag{3}$$

where, $\mathbf{u}$ is the velocity, $P$ is the dynamic pressure, $\eta$ is the viscosity, $\dot{\epsilon}$ is the strain rate tensor, $T$ is the temperature, $\mathbf{r}$ is the unit vector in radial direction, $B^{\mathrm{eff}}$ is the effective buoyancy ratio (defined below). $t$ is time, and $H$ is internal heating. $\xi = (R_e/D)^3$ with $R_e$ as the Earth's radius and $D$ as the mantle thickness. Physical parameters in the above equations are all non-dimensional. The Eqs. (1)–(3) are solved using the CitcomCU code, which is available at https://geodynamics.org/cig/software/citcomcu/.

The thermal Rayleigh number Ra is defined as:

$$Ra = \frac{\rho_0 g \alpha_0 \Delta T D^3}{\eta_0 \kappa_0} \tag{4}$$

where $\rho_0$, $\alpha_0$, $\Delta T$, $\eta_0$, $\kappa_0$ are dimensional reference values of background mantle reference density, thermal expansivity, temperature difference between core-mantle boundary and surface, reference viscosity at temperature $T = 0.6$ (non-dimensional), and thermal diffusivity, respectively. $g$ is dimensional gravitational acceleration.

The internal heating $H$ is non-dimensionalized as:

$$H = \frac{R_e^2}{\kappa_0 c_{P_0} \Delta T} H^* \tag{5}$$

where, $c_{P_0}$ is heat capacity, $H^*$ is the dimensional heat production rate.

The buoyancy number for a compositional component ($B_i$) is defined as the ratio between intrinsic density anomaly and density anomaly due to thermal expansion:

$$B_i = \frac{\Delta \rho_i}{\rho_0 \alpha_0 \Delta T} \tag{6}$$

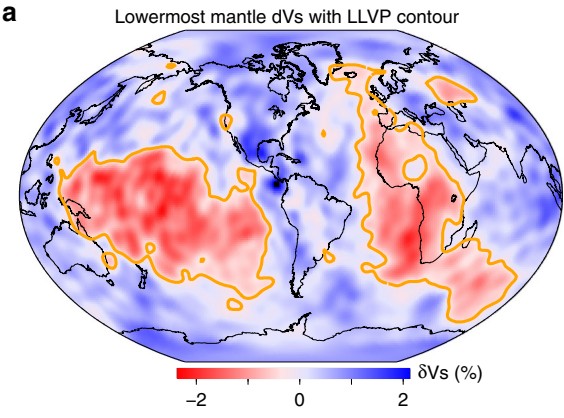

**a** Lowermost mantle dVs with LLVP contour

δVs (%)

−2   0   2

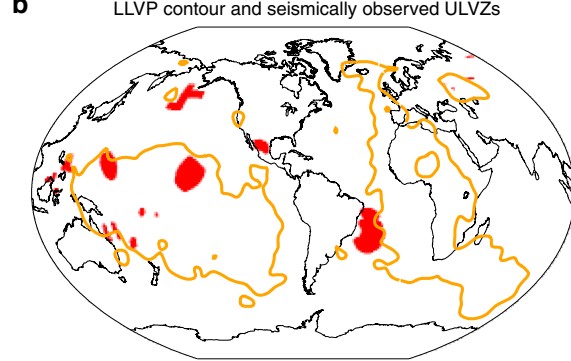

**b** LLVP contour and seismically observed ULVZs

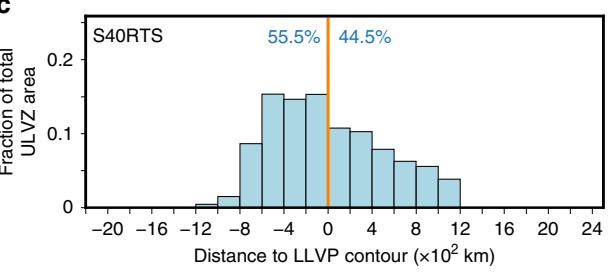

**c**

S40RTS    55.5%   44.5%

Fraction of total ULVZ area

Distance to LLVP contour (×10² km)

**Fig. 5** Seismic observation of ULVZs and LLVP edges. **a** Seismic shear-wave tomography model at 2800 km depth (S40RTS), with the LLVP edges shown by orange lines that surround 30% of the CMB area. **b** LLVP edges (*orange lines*) along with the ULVZ Fresnel zone patches (*red*, discretized in 0.5 × 0.5 degree cells) for all ULVZ waveform studies of core reflected waves (ScS, ScP, PcP). **c** The minimum distance of every ULVZ cell to LLVP edges. For each distance in the *horizontal axis*, the corresponding *vertical axis* shows the areal fraction of total ULVZ area. The *thick orange line* denotes the LLVP margin. Negative distance represents outside of LLVPs and positive distance represents inside of LLVPs. The percent area outside vs. inside of LLVPs is indicated by the *blue text*

where, $\Delta\rho_i$ is intrinsic density difference between an individual compositional component and the background mantle.

Similarly, the effective buoyancy ratio is defined as:

$$B^{\mathrm{eff}} = \frac{\Delta\rho_{\mathrm{el}}}{\rho_0\alpha_0\Delta T} \qquad (7)$$

where $\Delta\rho_{\mathrm{el}}$ is effective intrinsic density anomaly on an element computed from the intrinsic density anomaly and the fraction of each compositional component in the element using the hybrid tracer method described below.

**Hybrid tracer method.** In thermochemical geodynamical modeling, two methods are typically used to model the advection of compositional field: the ratio tracer method and absolute tracer method[37].

In the absolute tracer method, the composition fraction ($C_i$) of each compositional component (except the background mantle) is proportional to the

number of tracers per volume:

$$C_i = \frac{N_i V_0}{V} \qquad (8)$$

where $N_i$ is the number of tracers for the $i$th compositional component in an element, $V$ is the volume of the element and $V_0$ is a constant which equals to average volume per tracer for the $i$th compositional component. For background mantle, $N_i$ equals zero (i.e., no tracer in the element) and $C_i$ becomes zero. Thus, there is no need for additional tracers to simulate the background mantle. This becomes a big advantage when the volume of chemical heterogeneities is very small (e.g., the ULVZs), which could be efficiently simulated with a small amount of tracers.

For ratio tracer method, the background mantle is also represented by tracers. Usually, the density of the background mantle is the reference density and the buoyancy number for the background mantle equals zero. The compositional fraction ($C_i$) for each compositional component within an element is:

$$C_i = \frac{N_i}{N} \qquad (9)$$

where $N_i$ is the number of tracers in the element used to simulate the $i$th compositional component. $N$ is the total number of tracers in that element.

The ratio tracer method is benchmarked, and compared with absolute tracer method in ref. [37]. The ratio tracer method has several advantages over the absolute tracer method, such as minimal numerical diffusion and low entrainment. Thus, ratio tracer method is often used when dealing with large-scale chemical heterogeneities (i.e., LLVPs), because in this case the absolute tracer method also needs large amount of tracers and no longer has the advantage of modeling the compositional heterogeneities using less tracers.

In this study, our model is featured by both large-scale thermochemical piles and small-scale accumulations of ultradense material. We developed a hybrid tracer method which combines the advantages of ratio and absolute tracer method. Here, the background mantle and large scale compositional heterogeneities of piles are represented by ~710 million ratio tracers and the smaller scale accumulations of compositionally distinct ultradense material are simulated by ~52–110 million absolute traces (depending on the initial volume of ultradense material).

The effective intrinsic density anomaly ($\Delta\rho_{\mathrm{el}}$) for each element in the computation domain contains two parts. One part is from background mantle and pile material which are modeled with ratio tracers, and is given by:

$$\Delta\rho_{\mathrm{el}}^{\mathrm{r}} = \Delta\rho_{\mathrm{p}}C_{\mathrm{p}} + \Delta\rho_{\mathrm{bg}}C_{\mathrm{bg}} \qquad (10)$$

where, $\Delta\rho_{\mathrm{p}}$ is the intrinsic density anomaly of pile material. $C_{\mathrm{p}}$ is compositional fraction of pile material for the element which is calculated using Eq. (9). $\Delta\rho_{\mathrm{bg}}$ and $C_{\mathrm{bg}}$ are the intrinsic density anomaly and compositional fraction for the background mantle for the element, respectively. The intrinsic density anomaly of the background mantle is zero, so Eq. (10) becomes:

$$\Delta\rho_{\mathrm{el}}^{\mathrm{r}} = \Delta\rho_{\mathrm{p}}C_{\mathrm{p}} \qquad (11)$$

The other part of the effective intrinsic density anomaly on an element ($\Delta\rho_{\mathrm{el}}$) is from ultradense (i.e., ULVZ) material which is modeled with absolute tracers, and is given as:

$$\Delta\rho_{\mathrm{el}}^{\mathrm{a}} = \Delta\rho_{\mathrm{u}}C_{\mathrm{u}} \qquad (12)$$

where $\Delta\rho_{\mathrm{u}}$ is the intrinsic density anomaly of ultradense material. $C_{\mathrm{u}}$ is compositional fraction of ultradense material for the element which is calculated using Eq. (8). We truncated $C_{\mathrm{u}}$ at 1 to avoid unphysically settling of tracers[37].

In the hybrid tracer method, the effective intrinsic density anomaly on an element of the computational domain ($\Delta\rho_{\mathrm{el}}$) is given by:

$$\Delta\rho_{\mathrm{el}} = \Delta\rho_{\mathrm{el}}^{\mathrm{a}} + \Delta\rho_{\mathrm{el}}^{\mathrm{r}}(1 - C_{\mathrm{u}}) \qquad (13)$$

or,

$$\Delta\rho_{\mathrm{el}} = \Delta\rho_{\mathrm{u}}C_{\mathrm{u}} + \Delta\rho_{\mathrm{p}}C_{\mathrm{p}}(1 - C_{\mathrm{u}}) \qquad (14)$$

Notice that, for $C_{\mathrm{u}} = 0$ (element has no ultradense material), $\Delta\rho_{\mathrm{el}}$ is equivalently calculated using the ratio tracer method; for $C_{\mathrm{u}} = 1$ (element is saturated with ultradense material), $\Delta\rho_{\mathrm{el}}$ is equivalently calculated using the absolute tracer method.

The effective buoyancy ratio ($B^{\mathrm{eff}}$) on an element is related to the effective intrinsic density anomaly ($\Delta\rho_{\mathrm{el}}$) on this element by:

$$B^{\mathrm{eff}} = \Delta\rho_{\mathrm{el}}/(\rho_0\alpha_0\Delta T) \qquad (15)$$

**Core-reflected wave ULVZ studies.** In this study, we survey ULVZ studies that utilized core-reflected energy waveform analyses, e.g., PcP, ScP, and ScS. These

waves have the potential to detect ULVZ structure at the CMB reflection point location. This differs from studies using core waves, e.g., SPdKS which can have an uncertainty regarding mapping ULVZ structure at the core entrance or exit location of the path (similarly, PKP and PKKP have this ambiguity). Some Pdiff and Sdiff studies have evidence for ULVZs (e.g., refs. [7, 52]). While these analyses indicate specific ULVZ locations, the long paths of the diffracted wave result in some uncertainty as to where along the path the ULVZ is located. For this reason, we investigate ULVZ proximity to LLVP edges with just the core-reflected data. The studies, regions, and wave type are given in Supplementary Table 3.

Each of the ULVZ Fresnel zones of the studies in Supplementary Table 3 was decimated onto a 0.5 deg by 0.5 deg grid, with the area computed for each cell. The minimum distance to the nearest LLVP boundary is computed for each cell, and the fraction of the total ULVZ area summed up as a function of that minimum distance. This is display in Fig. 5c, as well as in Supplementary Fig. 11 for six tomographic models. The models are S40RTS[46], along with HMSL-S06[53], S362ANI[54], SEMUCB-WM1[55], SP12RTS[56], and GyPsum[57]. The LLVP boundary is chosen to be the contour that surrounds 30% of the CMB by area that has the lowest shear wave speeds in the tomography model[12]. The results for all the tomographic models are similar in that there is a significant area percentage of ULVZs located outside the LLVPs.

**Data availability**. The authors declare that all relevant data supporting the findings of this study are available within the article and its Supplementary Information file or available upon request. The code, CitcomCU, is available from https://geodynamics.org/cig/software/citcomcu/. The authors' specific version of the code is available upon request.

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

## Acknowledgements

We thank F. Deschamps, S. King and an anonymous reviwer for their constructive comments. The project is supported by NSF grants EAR-1045788, EAR-1401270 and EAR-1648817.

## Author contributions

M.L., A.K.M. and E.J.G. contributed to conceiving the idea. M.L. carried out the numerical calculation. A.K.M. supervised the project. S.Y. and E.J.G. reviewed previous studies on ULVZs and digitized the location and size of ULVZs. All authors contributed to writing the paper.

## Additional information

**Competing interests:** The authors declare no competing financial interests.

