## [Peer Review File · Nature Communications]

Reviewer #1 (Remarks to the Author)

1. What are the major claims of the paper?

The team of authors, well-known for leading the effort of our current understanding of the ULVZs, claim that these simulations indicate that the location and shapes of the ULVZs can be used to identify the mechanisms responsible for the formation of the ULVZs. Those observed near the center of LLVPs are likely thermal in origin, while those observed around the LLVP edges are likely compositional in nature.

2. Are they novel and will they be of interest to others in the community and the wider field?

The conclusion that ULVZs near the LLVP edges are not extremely new, as a 2010 article (ref 1) from this group had already demonstrated these results. One important distinction between that 2D work and this 3D model is the way ULVZs near LLVP edges extend into the third dimension. In the previous work, it was not possible to identify pinch-offs, which can now be done from this work.

3. If the conclusions are not original, it would be helpful if you could provide relevant references.

Is the work convincing, and if not, what further evidence would be required to strengthen the conclusions?

As I discuss above, I think the conclusions of the article are still quite original, and the insight provided by the new 3D models is extremely useful. I have one recommendation regarding comparing the results from their calculations with experimentally determined melting curves. I think it will strongly bolster their argument about the purely thermal ULVZs.

4. Do you feel that the paper will influence thinking in the field?

I think this paper will generate a wave of discussions in the field. Due to the cross-disciplinary nature, the impact will be felt beyond geodynamics and seismology and will include areas such as petrology, mineral physics, geochemistry, and planetary origins.

Major recommendation:

The models seem to be well-constrained, I have no problem with the technical details of the model. I will, however, like to see one additional task performed. Currently, the treatment of case 1 is a little vague. The authors ascribe the hottest 10% of the LLVP to the thermal ULVZs. I think it is worth dimensionalizing the least and the highest temperatures in this hottest area, and compare those temperatures to a realistic melting curve at the bottom of the mantle such as the one from ref (2).

The reason for doing such a comparison is twofold. 1. it will provide an additional constraint on the volume of melt in the ULVZs estimated by other workers, but more importantly, 2. it will cast some light on the stability of the melt. For example, if the dimensional temperature is close to the liquidus, then there is probably too much melt in these thermal ULVZs that can be retained over a geologically significant periods of time, and alternative explanations need to be invoked to explain the existence of these.

Finally, I'd suggest the authors do some editing for English. For example 'inboard' in line 101 and 103 is probably supposed to be 'inward'.

(1) McNamara, A. K., Garnero, E. J., & Rost, S. (2010). Tracking deep mantle reservoirs with ultra-low velocity zones. *Earth and Planetary Science Letters*, 299(1–2), 1–9.

(2) Fiquet, G., Auzende, a. L., Siebert, J., Corgne, A., Bureau, H., Ozawa, H., & Garbarino, G. (2010). Melting of Peridotite to 140 Gigapascals. *Science*, 329(5998), 1516–1518.

Reviewer #2 (Remarks to the Author)

In this study, the authors conducted two series of numerical experiments of thermo-chemical convection to better understand the possible nature of the ultra-low seismic velocity zones (ULVZ) observed at the bottom of the mantle. One series assumes that ULVZ are related to hottest regions within thermo-chemical piles, and the second one that they are related to patches of ultra-dense material that have migrated within thermo-chemical piles. The authors show that these two hypothesis lead to very different morphologies and distributions of ULVZ. According to their experiments, if ULVZ are related to hottest regions of piles, they would be located well within these piles. If, on the contrary, ULVZ are related to pockets of ultra-dense material, they would appear as discontinuous patches of various shapes, and would be concentrated mostly along the edges of the piles. This is an interesting article that brings new theoretical constraints from geodynamics on the nature of the ultra-low seismic velocity zones observed at the base of the mantle. Importantly, these predictions can be tested against existing or future seismic observations. In addition, the article is well structured and clearly written. I only have minor comments, which can be easily addressed by the authors. I therefore recommend this article for publication in Nature Communications after some minor revisions.

1. In the first set of experiments, the authors identify ULVZ with the hottest regions within the piles, and fix the boundary to the hottest 10% regions. This value sounds rather subjective. What motivated this choice? Also, it might be interesting that the authors rescale the non-dimensional temperature field they obtain, and compare this rescaled distributions with melting curve of pyrolite (e.g., that from Nomura et al., Science, 343, 522-525 (2014)). This of course is also a bit subjective, since it depends on the assumed super-adiabatic temperature jump, but still it would give some justification for the delimitation of the hottest regions.
2. Comparison between the morphology of ultra-dense structures and the observed ULVZ (lines 142-150): to follow the discussion, it would be helpful to add a map of the currently know ULVZ in supplementary material.
3. Systematic parameter study (lines 151-156). The authors have tested different values of controlling parameters, and show that the distribution of hottest regions and ultra-dense material is not substantially affected by changes in these controlling parameters. This is correct, but a closer examination at the different cases (supplementary figures S2-S9) shows strong differences in the morphology of these regions. This deserves a short discussion. For instance, compared to the case shown in figure 2, increasing the sensitivity of viscosity with temperature (which, I would consider as likely in the case of the Earth's mantle) lead to much larger ultra-dense patches (Figure S5). Also, the authors consider the case of ultra-dense material intrinsically less viscous than the regular mantle material, but not the opposite case. Why? Are there any reasons that would justify this?
4. Origin and initial migration of ultra-dense material (line 163-170). The authors state that the presence of ULVZ outside piles of dense material may be due to the fact that these ULVZ didn't achieve their migration towards the piles. It would be good to show a series of plot (or a movie) showing initial migration in supplementary material. Also, this migration seems to be rather quick. The authors assume an initial condition consisting in a thin ubiquitous layer of ultra-dense material, but in the case of Earth mantle, a mechanism producing ultra-dense material regularly (interaction with outer core?) would be required to explain the presence of ULVZ outside piles (LLVPs) today. More generally, the possible origins of the ultra-dense material (remnant of magmatic ocean, outer core percolation, ...) is not clearly discussed in the manuscript, and the authors may add a few lines on this in their revised manuscript.
5. In introductory part (lines 38-51), the authors may mention a recent study by Li et al., GRL, 43, 3215-3225 (2016), showing that patches of pPv may temporarily be stable within piles of dense material, providing another possible explanation for ULVZ (I apologize for citing this paper on which I am co-author, but it is really relevant in the context of the authors' study).
6. Three typos - Line 106. Add a "s" to consider - Line 164. Replace "ultra-denes" with "ultra-dense". - Legend of Figure S4. "in addition to temperature dependent viscosity" is indicated twice. Frederic Deschamps

Reviewer #3 (Remarks to the Author)

The manuscript "Compositionally-distinct ultra-low velocity zones on Earth's core-mantle boundary" by Li and McNamara addresses the question of whether the observed ULVZs at the base of Earth's mantle are more likely caused by melting, which would imply the hottest regions of the LLSVZs or composition. In this regard figure 4 is compelling, showing a distinct difference between the location of the ULVZs with respect to the LLSVZs. I recommend publishing after a few minor revisions. I do not need to see a revised version of this manuscript.

The work here is clearly explained and carefully presented. All necessary materials to understand the details of the models are carefully tucked away in the supplemental materials.

There is one thing that would make the contribution stronger. If the authors drafted a figure like figure 4 based on the observed locations of ULVZs in relationship to the observed LLSVZs (in say seismic model S40RTS) this would connect the modeling work to the observations. In the text the authors make the statement that the computational pattern is a better fit to the observations, yet they do not show us this to be the case. I fully expect that the observations will be messier than the models, yet the distinction between the two hypotheses in the paper are so strong that I would expect even with uncertainties the pattern should be observed.

Minor:

Lines 61-62: Given that Citcom has traditionally used a Radius scaling for the Rayleigh number, not a depth scaling it would be worth a short sentence in the main body of the text to clarify. It would keep the reader who is savvy to such destinations from having to flip back to the supplemental material to check. I appreciate your use of the form of non-dimensionalization from Chandrasaekar which keeps the original definition of the Rayleigh number. This is important, otherwise this Rayleigh number looks very low.

It is not clear what the average radially-averaged viscosity looks like for these models. This would be useful to see in the online supplement. It would help the reader see something about the convective vigor of the calculations.

Line 96-97: "depending on choices of reference temperature and thermal expansivity". I think you mean "based on the choice of reference temperature and thermal expansivity". This also appears somewhere in the supplemental material. I did not flag the location.

Reference 9 seems rather incomplete. More like a note to self. Also, is this really the best reference to Tackley's group's work on this?

Best Regards and Nice Work, Scott King

Response to Reviewer #1:

1. What are the major claims of the paper?

The team of authors, well-known for leading the effort of our current understanding of the ULVZs, claim that these simulations indicate that the location and shapes of the ULVZs can be used to identify the mechanisms responsible for the formation of the ULVZs. Those observed near the center of LLVPs are likely thermal in origin, while those observed around the LLVP edges are likely compositional in nature.

2. Are they novel and will they be of interest to others in the community and the wider field?

The conclusion that ULVZs near the LLVP edges are not extremely new, as a 2010 article (ref 1) from this group had already demonstrated these results. One important distinction between that 2D work and this 3D model is the way ULVZs near LLVP edges extend into the third dimension. In the previous work, it was not possible to identify pinch-offs, which can now be done from this work.

3. If the conclusions are not original, it would be helpful if you could provide relevant references. Is the work convincing, and if not, what further evidence would be required to strengthen the conclusions?

As I discuss above, I think the conclusions of the article are still quite original, and the insight provided by the new 3D models is extremely useful. I have one recommendation regarding comparing the results from their calculations with experimentally determined melting curves. I think it will strongly bolster their argument about the purely thermal ULVZs.

4. Do you feel that the paper will influence thinking in the field?

I think this paper will generate a wave of discussions in the field. Due to the cross-disciplinary nature, the impact will be felt beyond geodynamics and seismology and will include areas such as petrology, mineral physics, geochemistry, and planetary origins.

Response: Thanks for the very positive comments.

Major recommendation:

The models seem to be well-constrained, I have no problem with the technical details of the model. I will, however, like to see one additional task performed. Currently, the treatment of case I is a little vague. The authors ascribe the hottest 10% of the LLVP to the thermal ULVZs. I think it is worth dimensionalizing the least and the highest temperatures in this hottest area, and compare those temperatures to a realistic melting curve at the bottom of the mantle such as the one from ref (2) (Fiquet et al., 2010). The reason for doing such a comparison is twofold. 1. it will provide an additional constraint on the volume of melt in the ULVZs estimated by other workers, but more importantly, 2. it will cast some light on the stability of the melt. For example, if the dimensional temperature is close to the liquidus, then there is probably too much melt in these thermal ULVZs that can be retained over a geologically significant periods of time, and alternative explanations need to be invoked to explain the existence of these.

Response: Thanks for this very good suggestion. We have dimensionalized the temperature of the hottest 10% regions and compare them with the solidus and liquidus temperature on the

CMB that have been constrained by mineral physical experiments from different studies. Through this comparison, we show that the hottest 10% regions are candidate locations for partial melting. The additional texts are highlighted in red in lines 128-137. However, before doing this comparison, we discuss in lines 98-119 that there are large uncertainties for the solidus and liquidus temperature at CMB pressure, largely because the lowermost mantle composition is not well constrained. For example, we don't know how much H₂O exists in the lowermost mantle, but previous mineral physics studies showed that H₂O has a significant effect on the melting temperature. In addition, dimensionalization of temperature in models is directly dependent on CMB temperature, which is not well constrained by observations. Because of these uncertainties, we consider it impractical to state with certainty the amount of partial melting above the CMB in our models, by comparing the lowermost mantle temperature in our models with the solidus and liquidus temperatures from mineral physics studies. Therefore, in light of these uncertainties, we mostly focus on searching the hottest regions in the lowermost mantle to provide the best candidate regions for partial melting.

Given the uncertainties, we agree with the reviewer that this would help readers better understand the temperature of the hottest regions and would help them get some sense of the stability of the melts. We thank the reviewer for this very good suggestion.

Finally, I'd suggest the authors do some editing for English. For example 'inboard' in line 101 and 103 is probably supposed to be 'inward'.

Response: Thanks for the suggestion. We have replaced the word 'inboard' with 'inward' (line 128) or other clearer descriptions (line 139).

Response to Reviewer #2:

In this study, the authors conducted two series of numerical experiments of thermo-chemical convection to better understand the possible nature of the ultra-low seismic velocity zones (ULVZ) observed at the bottom of the mantle. One series assumes that ULVZ are related to hottest regions within thermo-chemical piles, and the second one that they are related to patches of ultra-dense material that have migrated within thermo-chemical piles. The authors show that these two hypothesis lead to very different morphologies and distributions of ULVZ. According to their experiments, if ULVZ are related to hottest regions of piles, they would be located well within these piles. If, on the contrary, ULVZ are related to pockets of ultra-dense material, they would appear as discontinuous patches of various shapes, and would be concentrated mostly along the edges of the piles.

This is an interesting article that brings new theoretical constraints from geodynamics on the nature of the ultra-low seismic velocity zones observed at the base of the mantle. Importantly, these predictions can be tested against existing or future seismic observations. In addition, the article is well structured and clearly written. I only have minor comments, which can be easily addressed by the authors. I therefore recommend this article for publication in Nature Communications after some minor revisions.

Response: Thanks for the very positive comments.

1. In the first set of experiments, the authors identify ULVZ with the hottest regions within the piles, and fix the boundary to the hottest 10% regions. This value sounds rather subjective. What motivated this choice? Also, it might be interesting that the authors rescale the non-dimensional temperature field they obtain, and compare this rescaled distributions with melting curve of pyrolite (e.g., that from Nomura et al., Science, 343, 522-525 (2014)). This of course is also a bit subjective, since it depends on the assumed super-adiabatic temperature jump, but still it would give some justification for the delimitation of the hottest regions.

Response: Thanks for this very good suggestion. The same comment has also been raised by the first reviewer. In our first case, we examine the locations of ULVZs that are caused by partial melting alone. However, we consider it impractical to determine the amount of partial melting above the CMB in our models, by comparing the lowermost mantle temperature with the solidus and liquidus temperatures from mineral physics studies. The reasons are: firstly, there are large uncertainties of solidus temperature at the CMB pressure. For example, the solidus temperature is strongly affected by the content of H₂O, and it is unclear how much H₂O exists in the lowermost mantle. Secondly, the lowermost mantle temperature is not well constrained. There is about 1,000 K of uncertainty for the CMB temperature. Thirdly, our models do not constrain the lowermost mantle temperature and we agree with the reviewer that it depends on our choice of reference temperature for dimensionalization. We have discussed these uncertainties in the revised manuscript (highlighted in red in lines 98-119). In light of these uncertainties, we mostly focus on searching the hottest regions in the lowermost mantle to provide the best candidate regions for partial melting.

Although there are uncertainties, we agree with the reviewer that by changing the dimensionless temperature into dimensional values, we could give some justification for the delimitation of the hottest regions. We thus dimensionalized the temperature of the hottest 10% regions, and

compare it with the solidus and liquidus temperatures constrained from 3 different groups of mineral physics studies. Through this comparison, we show that the hottest 10% regions are candidate regions for partial melting. The related texts are highlighted in lines 128-137.

2. Comparison between the morphology of ultra-dense structures and the observed ULVZ (lines 142-150): to follow the discussion, it would be helpful to add a map of the currently know ULVZ in supplementary material.

Response: Thank for this very good suggestion. We have added a new figure (Figure 5) to show the seismic observation of ULVZs. We also calculate the distances between ULVZs and LLVP edges and compare them with our numerical modeling studies.

In Figure 5a, we show the seismic tomography model near the CMB from S40RTS. In Figure 5b, we show the seismic observations of ULVZs. However, we only show ULVZs whose location and size are well determined with the least amount of uncertainty: from core reflected wave studies. In Figure 5c, we show the distances from ULVZs to the edges of LLVPs. We also calculate the distance between ULVZs and the LLVP edges that are defined in other 5 different seismic tomography models. This is shown in Supplementary Information Figure 11. Our results show that observed ULVZs indeed are dominantly located near LLVP edges. The is similar to the geodynamically predicted distribution of ultra-dense material that collects at the edges of thermochemical piles.

We have added a new section ‘comparison with seismic observations of ULVZs’ to the revised manuscript, which is motivated by the reviewer. We think this section significantly improves our paper, and we appreciate this suggestion from the reviewer.

And as we mentioned, we have added a new coauthor Shule Yu, whose contribution was instrumental in computing ULVZ area and distances to LLVPs, and hence key for constructing Figure 5 and supplementary Figure 11, as well as the supplementary Table 3, of the ULVZ studies utilized.

3. Systematic parameter study (lines 151-156). The authors have tested different values of controlling parameters, and show that the distribution of hottest regions and ultra-dense material is not substantially affected by changes in these controlling parameters. This is correct, but a closer examination at the different cases (supplementary figures S2-S9) shows strong differences in the morphology of these regions. This deserves a short discussion. For instance, compared to the case shown in figure 2, increasing the sensitivity of viscosity with temperature (which, I would consider as likely in the case of the Earth’s mantle) lead to much larger ultra-dense patches (Figure S5). Also, the authors consider the case of ultra-dense material intrinsically less viscous than the regular mantle material, but not the opposite case. Why? Are there any reasons that would justify this?

Response: Thanks for this suggestion. We have added a section to briefly discuss the results of all cases that are shown in the supplementary information. The discussion is highlighted in red in lines 199-217. A more detailed discussion about the modeling results for these cases is provided in the Supplementary Information.

Several previous studies have suggested that the ULVZs are enriched in iron. The Fe-enriched ULVZs may have low solidus temperature and may be partially molten. If so, the viscosity of these ULVZs is expected to be reduced. So, we test one case in which the ultra-dense material has intrinsically lower viscosity than the regular mantle material. In doing so, we mainly wanted to explore a viscosity reduction due to possible partial melting, but we don't envision a typical scenario in which ULVZs would have a higher intrinsic viscosity than the surroundings. It cannot be ruled out that ULVZs could have significantly larger grain-size (relative to surroundings) which would act to increase their diffusion creep viscosity. However, the associated viscosity increase would have to overcome the strong viscosity decrease due to temperature-dependence. In any case, we consider this possibility to be beyond the scope of the present manuscript. This part of discussion is included in the Supplementary Information.

4. Origin and initial migration of ultra-dense material (line 163-170). The authors state that the presence of ULVZ outside piles of dense material may be due to the fact that these ULVZ didn't achieve their migration towards the piles. It would be good to show a series of plot (or a movie) showing initial migration in supplementary material. Also, this migration seems to be rather quick. The authors assume an initial condition consisting in a thin ubiquitous layer of ultra-dense material, but in the case of Earth mantle, a mechanism producing ultra-dense material regularly (interaction with outer core?) would be required to explain the presence of ULVZ outside piles (LLVPs) today. More generally, the possible origins of the ultra-dense material (remnant of magmatic ocean, outer core percolation, ...) is not clearly discussed in the manuscript, and the authors may add a few lines on this in their revised manuscript.

Response: Thanks for this suggestion. We have provided two movies in the supplementary information to show the evolution of distribution of hottest regions and ultra-dense material in the lowermost mantle. Supplementary Movie 1 shows that the hottest regions always occur well within the thermochemical piles throughout the calculation for Case 1. Supplementary Movie 2 illustrates that the initial global layer of ultra-dense material is advected to the edges of thermochemical piles, and forms into discontinuous patches with variable size and shape along the pile edges. We also select some snapshots from supplementary Movie 2 to show the initial migration of ultra-dense material from outside of piles to at the edges of piles.

We have added discussion on the possible origins of the ultra-dense material in the revised manuscript in lines 263-277. The discussion is also described below:

For the Earth, the crystallization of basal magma oceanic may initially produce a thin layer of ultra-dense material on the CMB. The ultra-dense materials may be produced by the interaction between the core and the mantle and they may be produced at any location where the core and mantle interacts. In addition, the subduction of slabs may bring some intrinsically dense materials to the lowermost mantle outside of the LLVPs. Our experiments are geared toward understanding thermochemical convection at equilibrium conditions, however, our model setup also allows us to explore (in a limited manner) the how ultra-dense material gets swept from the surrounding mantle to the edges of thermochemical piles. Because our initial condition consists of a thin, uniform ultra-dense ubiquitous layer along the CMB, the early times of the calculation exhibit the sweeping of this material toward the piles (Supplementary Movie 2, Supplementary

Fig. 12). It demonstrates that any high density compositional heterogeneity outside of piles is being advected toward the global upwelling regions (where the piles exist). Therefore, if ULVZs are caused by ultra-dense subduction remnants or core-mantle boundary reaction products, we expect to observe them outside of piles as they are being advected toward them.

5. In introductory part (lines 38-51), the authors may mention a recent study by Li et al., GRL, 43, 3215-3225 (2016), showing that patches of pPv may temporarily be stable within piles of dense material, providing another possible explanation for ULVZ (I apologize for citing this paper on which I am co-author, but it is really relevant in the context of the authors' study).

Response: We agree that this work is very relevant to our study. Thanks for pointing this out. We have mentioned this study in the introduction of the revised manuscript (lines 43-44).

6. Three typos

- Line 106. Add a "s" to consider

- Line 164. Replace "ultra-denes" with "ultra-dense".

- Legend of Figure S4. "in addition to temperature dependent viscosity" is indicated twice.

Response: Thanks. We have corrected these typos.

Response to Reviewer #3:

The manuscript "Compositionally-distinct ultra-low velocity zones on Earth's core-mantle boundary" by Li and McNamara addresses the question of whether the observed ULVZs at the base of Earth's mantle are more likely caused by melting, which would imply the hottest regions of the LLSVZs or composition. In this regard figure 4 is compelling, showing a distinct difference between the location of the ULVZs with respect to the LLSVZs. I recommend publishing after a few minor revisions. I do not need to see a revised version of this manuscript. The work here is clearly explained and carefully presented. All necessary materials to understand the details of the models are carefully tucked away in the supplemental materials.

Response: Thanks for the very positive comments.

There is one thing that would make the contribution stronger. If the authors drafted a figure like figure 4 based on the observed locations of ULVZs in relationship to the observed LLSVZs (in say seismic model S40RTS) this would connect the modeling work to the observations. In the text the authors make the statement that the computational pattern is a better fit to the observations, yet they do not show us this to be the case. I fully expect that the observations will be messier than the models, yet the distinction between the two hypotheses in the paper are so strong that I would expect even with uncertainties the pattern should be observed.

Response: Thanks for this very good suggestion, which was also mentioned by Reviewer 2. As we mentioned, we have added a new figure (Figure 5) to show observations of LLVPs and ULVZs and the distances between ULVZs and the edges of LLVPs. The results are compared with our geodynamic modeling results.

Lines 61-62: Given that Citcom has traditionally used a Radius scaling for the Rayleigh number, not a depth scaling it would be worth a short sentence in the main body of the text to clarify. It would keep the reader who is savvy to such destinations from having to flip back to the supplemental material to check. I appreciate your use of the form of non-dimensionalization from Chandrasekar which keeps the original definition of the Rayleigh number. This is important, otherwise this Rayleigh number looks very low.

Response: We agree with the reviewer. We have further clarified that we use the mantle thickness to compute the Rayleigh number (highlighted in red in lines 64-65).

It is not clear what the average radially-averaged viscosity looks like for these models. This would be useful to see in the online supplement. It would help the reader see something about the convective vigor of the calculations.

Response: Thanks for pointing this out. We have provided the average viscosity in the Supplementary Figure 1.

Line 96-97: "depending on choices of reference temperature and thermal expansivity". I think you mean "based on the choice of reference temperature and thermal expansivity". This also appears somewhere in the supplemental material. I did not flag the location.

Response: Thanks for the comments. We have provided the reference parameters used in this study in Supplementary Table 2.

Reference 9 seems rather incomplete. More like a note to self. Also, is this really the best reference to Tackley's group's work on this?

Response: We have removed this reference.

Reviewers' Comments:

Reviewer #1:

Remarks to the Author:

I am satisfied with the work done by the team in order to answer my question regarding melting. The new text discussing the uncertainties associated with the CMB temperature and the hydrous peridotite solidus temperature is appropriate. Just a passing comment, though, the presence of high density phases, likely caused by the presence of Fe, can also reduce the melting temperature, influence the stability fields of the solid phases, and reduce viscosity due to compositional effects. That said, I don't think these issues need to be addressed in this manuscript.

Before acceptance, please run another check on the typos. This time I found 'outboard' instead of 'outward'. After this check, I think the manuscript is ready for publication.

Reviewer #2:

Remarks to the Author:

I have read the revised version of this article, and found that the authors have taken into account all suggestions and answered all questions raised by myself and the other reviewers.

I only have one very small question regarding supplementary movie 2 : at around time $t = 8s$, we can see some ultra-dense material appearing somewhere in the middle of the LLVP, but the authors mentioned that all ultra-dense material is initially distributed in a thin basal layer. Where does the ultra-dense material appearing at $t = 8s$ come from ? Does it correspond to small fraction of ultra-dense material that have accumulated along the edge of the LLVP and have been entrained by the flow and recycled ?

Once the authors have clarified this very minor point, I recommend this manuscript for publication to Nature Communication in its present form.

Response to Reviewer #1:

I am satisfied with the work done by the team in order to answer my question regarding melting. The new text discussing the uncertainties associated with the CMB temperature and the hydrous peridotite solidus temperature is appropriate. Just a passing comment, though, the presence of high density phases, likely caused by the presence of Fe, can also reduce the melting temperature, influence the stability fields of the solid phases, and reduce viscosity due to compositional effects. That said, I don't think these issues need to be addressed in this manuscript. Before acceptance, please run another check on the typos. This time I found 'outboard' instead of 'outward'. After this check, I think the manuscript is ready for publication.

Response: Thanks for the positive comments. We have fixed typos in the manuscript.

Response to Reviewer #2:

I have read the revised version of this article, and found that the authors have taken into accounts all suggestions and answered all questions raised by myself and the other reviewers. I only have one very small question regarding supplementary movie 2 : at around time $t = 8s$, we can see some ultra-dense material appearing somewhere in the middle of the LLVP, but the authors mentioned that all ultra-dense material is initially distributed in a thin basal layer. Where does the ultra-dense material appearing at $t = 8s$ come from ? Does it correspond to small fraction of ultra-dense material that have accumulated along the edge of the LLVP and have been entrained by the flow and recycled ? Once the authors have clarified this very minor point, I recommend this manuscript for publication to Nature Communication in its present form.

Response: Thanks for the positive comments. Yes, some ultra-dense material is entrained into thermochemical piles by mantle flow. This phenomenon is also shown in Figure 3b and Figure 3e in which the ultra-dense material that is entrained into piles is pointed to by the yellow arrow. The entrainment process is also described in Figure 4 and the U4 in Figure 4c is caused by stirring of ultra-dense material with pile material.